# Molecular Events behind the Selectivity and Inactivation Properties of Model NaK-Derived Ion Channels

**DOI:** 10.3390/ijms23169246

**Published:** 2022-08-17

**Authors:** Ana Marcela Giudici, María Lourdes Renart, Ana Coutinho, Andrés Morales, José Manuel González-Ros, José Antonio Poveda

**Affiliations:** 1Instituto de Investigación, Desarrollo e Innovación en Biotecnología Sanitaria de Elche (IDiBE), and Instituto de Biología Molecular y Celular (IBMC), Universidad Miguel Hernández, 03202 Elche, Spain; 2iBB-Institute for Bioengineering and Bioscience, Instituto Superior Técnico, Universidade de Lisboa, 1049-001 Lisboa, Portugal; 3Departamento de Fisiología, Genética y Microbiología, Universidad de Alicante, 03080 Alicante, Spain

**Keywords:** potassium channels, selectivity, inactivation, conformational flexibility, ion binding, thermal stability, homo-FRET, anisotropy decays, time-resolved and steady-state anisotropy

## Abstract

Y55W mutants of non-selective NaK and partly K^+^-selective NaK2K channels have been used to explore the conformational dynamics at the pore region of these channels as they interact with either Na^+^ or K^+^. A major conclusion is that these channels exhibit a remarkable pore conformational flexibility. Homo-FRET measurements reveal a large change in W55–W55 intersubunit distances, enabling the selectivity filter (SF) to admit different species, thus, favoring poor or no selectivity. Depending on the cation, these channels exhibit wide-open conformations of the SF in Na^+^, or tight induced-fit conformations in K^+^, most favored in the four binding sites containing NaK2K channels. Such conformational flexibility seems to arise from an altered pattern of restricting interactions between the SF and the protein scaffold behind it. Additionally, binding experiments provide clues to explain such poor selectivity. Compared to the K^+^-selective KcsA channel, these channels lack a high affinity K^+^ binding component and do not collapse in Na^+^. Thus, they cannot properly select K^+^ over competing cations, nor reject Na^+^ by collapsing, as K^+^-selective channels do. Finally, these channels do not show C-type inactivation, likely because their submillimolar K^+^ binding affinities prevent an efficient K^+^ loss from their SF, thus favoring permanently open channel states.

## 1. Introduction

During the last decades, structural studies on prokaryotic ion channels have been essential to understanding the molecular level function of these important membrane proteins and their analogous eukaryotic counterparts. The relative ease of the heterologous expression and purification of these channels allowed for the characterization of the conformational landscape of a variety of Na^+^, K^+^, and non-selective channels by different biophysical approaches. In particular, KcsA, a K^+^-selective prokaryotic channel from *S. lividans*, was the first ion channel to be crystallized, and its structure was solved at high resolution by X-ray diffraction methods [1,2]. The available structural information sheds light on the molecular mechanisms of the main properties of such proteins: ion permeation at high rates, gating, selectivity, and inactivation. Thus, the conduction of K^+^ at high rates has been associated with a “knock-on” mechanism that requires electrostatic repulsions of the permeating species within the pore [3]. On the other hand, the modulation of the permeation process through a slow C-type inactivation has been associated with a loss of K^+^ (ion depletion) at the selectivity filter (SF) [4,5], which explains why C-type inactivation in eukaryotic channels is favored at low K^+^ concentrations [6,7,8,9], while the presence of K^+^ inside the SF (“foot in the door” model) stabilizes the conductive conformation [10,11,12]. Furthermore, the selectivity properties of SF have been associated with the number of ion binding sites and the chemical properties of the amino acids forming the signature sequence [13].

One of the more recently crystallized prokaryotic channels has been the non-selective NaK channel, cloned from *Bacillus cereus.* The overall structure of this membrane protein is very much alike other prokaryotic K^+^ channels: four identical subunits arranged around a central pore, each one defined by two transmembrane helices (TM1 and TM2) connected by an extracellular loop, an amphiphilic N-terminal domain, and a cytoplasmic C-terminal domain [14]. The external loop contains the pore-helix and the SF. This SF domain shows the most significant difference with respect to the K^+^ selective channels, as it contains only two consecutive K^+^ binding sites in the signature sequence, named S3 and S4, followed by an external vestibule (Figure 1). This is in contrast to the four consecutive K^+^ binding sites (S1 to S4) found in the K^+^ selective proteins. Both closed (full length) and open (Δ19) states of the NaK channel have been successfully solved in the presence of several cations, and the first results described an almost identical SF conformation in the presence of either Na^+^ or K^+^ [13,14,15,16]. This lack of conformational plasticity was, in fact, claimed as the basis for the lack of selectivity in this protein, which presents a K^+^/Na^+^ selectivity ratio near 1 [13]. In contrast to such a proposal, more recent nuclear magnetic resonance spectroscopy (NMR), cryoelectron microscopy (cryoEM), and molecular dynamics (MD) approaches reported that the SF of the NaK channel was not static at all and that, indeed, the high conformational dynamics of the pore was the basis for the lack of selectivity in this channel protein [17,18,19,20]. 

The NaK channel is also a valuable tool for studying the influence of the number of binding sites within the SF on the functional properties of the protein. By a series of punctual mutations in the signature sequence, the two-K^+^-binding site SF could be transformed into a three- or four-binding site SF in which K^+^ selectivity is progressively enhanced [13]. Even though the four-K^+^-binding site NaK2K mutant presents an SF signature sequence identical to that of KcsA (Figure 1), its functional properties present important differences with respect to this latter channel. Thus, whereas KcsA presents a high K^+^/Na^+^ selectivity ratio (between 100–1000) [21,22,23] and a low open probability due to the presence of a C-type inactivation process [8,24], the NaK2K channel also conducts Na^+^ in the absence of K^+^ and no inactivation process has been reported so far [13]. Altogether, this indicates that, although the increase in the number of binding sites within the SF and their amino acid sequence are important to define the selectivity properties in ion channels, a more complex modulation, possibly mediated by interactions with nearby structures behind the SF, should also play an essential role in the functional properties of these channel proteins (Figure 1).

In previous studies, we have reported the conformational dynamics of the K^+^-selective channel KcsA through the application of intrinsic fluorescence-monitored thermal denaturation techniques combined with advanced homo-Förster Resonance Energy Transfer (homo-FRET) methodologies [25,26]. In particular, homo-FRET is a distance-dependent photophysical process where energy is transferred between two identical fluorophores via a long-range dipole–dipole coupling mechanism [27]. The energy transfer rate constant is inversely related to the sixth power of the distance separating the two fluorophores, and so, hetero- and homo-FRET measurements have been widely used as a spectroscopic molecular ruler [28]. Indeed, the changes in homo-FRET efficiency are currently being used to calculate intramolecular and intermolecular interactions (including conformational changes and protein oligomerization), not only in solution but also in live cells through the implementation of fluorescence anisotropy imaging microscopy (FAIM) [29,30]. In the W67 KcsA mutant channel, we found that the SF interacts with permeant and blocking cations in different ways, presenting distinct degrees of conformational plasticity and different affinities for each group of cations [25,26]. These properties were proposed to play an important role in defining the selectivity and inactivation process in this channel. Here, we have used Y55W mutants of the NaK and NaK2K channels, where the position of the introduced Trp residue is equivalent to that in the W67 KcsA mutant channel [25]. Thermal denaturation assays, steady-state fluorescence, and time-resolved homo-FRET analysis have been carried out in detergent micelles of these proteins, and their functional behavior was assessed upon their reconstitution in asolectin liposomes. The exploration of these Trp mutants has succeeded in characterizing the ion-protein interactions in these channels and in explaining its consequences on their channel properties, in which the large conformational flexibility detected at their SFs appears as a major modulatory factor.

## 2. Results

### 2.1. Functional Behavior of Y55W Mutants of Purified NaK and NaK2K Channels Reconstituted into Asolectin Giant Liposomes

Because most of our experiments required fluorescence monitoring, we introduced a tryptophane at position 55 of these channels in substitution of the native tyrosine. Such a position is equivalent to that of W67 in the potassium-selective KcsA channel, which we previously used as a reporter fluorescence group to study the interaction of cations with the KcsA selectivity filter [25]. The consequences of such a mutation on the most relevant functional features of such mutant NaK-derived channels, particularly their cation selectivity properties, have been tested by patch-clamp measurements on excised patches from reconstituted asolectin giant liposomes (Figure 1). Currently, there is no knowledge about any stimulus to open these channels, and for this reason, the W55Y mutants of the NaK and NaK2K channels used in this work also include the deletion of the first nineteen residues at the N-terminal of the subunits. This facilitates the opening of the channel’s inner gate, as confirmed by X-ray crystallography, and thus ion flow [13,15].

Under our reconstitution conditions (asolectin as the lipid matrix and a moderately high protein to lipid ratio), both NaK and NaK2K mutant channels exhibited activity, both at the macroscopic (Appendix A) and at the single-channel level (Figure 1). Macroscopic ion flux was measured through a fluorescence-based assay previously used with this and other ion channels [31] and shows that NaK2K allows a higher flux of K^+^ than NaK. Moreover, patch-clamp recordings often exhibit large Na^+^ or K^+^ currents, amenable for electrophysiological characterization at the single-channel level, without the need for the F92A mutation used by other groups for current enhancement [13,14,15,32,33]. According to a previous report, F92 presents two different conformers in equilibrium, which either allow or prevent K^+^ passage [34]. Thus, given our experimental results, it is reasonable to assume that our reconstitution conditions somehow favor the F92 rotameric form, allowing potassium flow, making the F92A mutation unnecessary. Still, it seems that the small currents reported by others in the absence of the F92A mutation might be present in some of our recordings superimposed on top of the currents mentioned above (see left panel in Figure 1). Furthermore, as detected in KcsA [35], the large size of the observed currents and the complexity of the recordings suggest the possible occurrence of clustering in these channels, perhaps leading to the coexistence of different clustered channel species in the reconstituted liposomes. This could be a consequence of the lipid-to-protein ratio used in this work, although samples prepared at different lipid-to-protein residues exhibited similar behavior. In any case, we did not consider this issue relevant to our purposes and did not pursue it any further.

From the continuous recordings at symmetrical conditions of either Na^+^ or K^+^, at positive and negative potentials, some differences between both channels are appreciated. It is important to note that the intracellular channel blocker TPeA (tetrapentyl ammonium) was added to the bath to ensure that recorded currents are from the channels oriented with their external side facing the bath solution; that is, an outside-out configuration. Although the open probability (Po) is quite high at negative voltages in all cases, at positive voltages, the activity was more flickering, particularly in the NaK2K, which showed a much lower Po. The conductance was also lower in this case (see left panel in Figure 1), revealing a marked inward rectification, mostly in NaK2K. Another interesting fact is that NaK shows a similar conductance for both ions in either condition, while NaK2K has a very high conductance for K^+^ at negative voltages, almost double that for Na^+^ in this channel or that for either cation in NaK. This suggests that NaK2K has a higher selectivity for K^+^ vs. Na^+^, in contrast to NaK, which showed a similar conductance for both cations. To quantify this selectivity, we measured the reversal potential values under competing conditions, which yields a more accurate estimation of this parameter, as these values do not depend on absolute ion concentrations but only on ion concentration ratios. In these experiments, we placed 200 mM K^+^ in the bath solution and a 180 mM concentration of Na^+^, along with 20 mM of K^+^, in the pipette solution. The current reversal potential was then introduced in the Goldman–Hodgkin–Katz equation (Equation (1) in Section 4) to calculate the corresponding permeability ratio relative to K^+^. In the NaK channel, the reversal potential was around 0 mV, which corresponds to a Na^+^/K^+^ permeability ratio of 1.0 and indicates that this channel is truly non-selective for Na^+^ versus K^+^. On the contrary, the NaK2K channel exhibited a reversal potential of +18.6 ± 0.4 mV (mean ± S.D., *n* = 3), yielding a Na^+^/K^+^ permeability ratio of 0.45 ± 0.01 (mean ± S.D., *n* = 3), meaning that the NaK2K channel selects moderately well K^+^ over Na^+^ under our experimental conditions. Nonetheless, a higher selectivity for K^+^ has been reported previously for the F92A mutation of this channel reconstituted at much higher lipid-to-protein ratios and in a different lipid media (POPE/POPG, 3:1 by weight) [33].

Another salient feature of these channels is their apparent lack of C-type inactivation. This was somehow expected since X-ray and NMR studies [13,14,15,36] did not detect inactivated-like structures under any experimental condition. This seems to be confirmed by our patch-clamp studies in which long, continuous recordings of up to several minutes duration showed maintained channel activity from the two channels, unaffected by time (data not shown).

From all these results, it follows that the higher ion flux of NaK2K relative to NaK detected at the macroscopic level would be caused by its higher conductance for K^+^ and not by a lower open probability or a C-type inactivation process in NaK.

### 2.2. Fluorescent Properties of Y55W Mutants of NaK and NaK2K Channels in DDM Micelles

Previous studies in the KcsA channel have shown that the fluorescence emitted by W67, located at the pore helix, nicely reports on cation occupancy-induced conformational changes in the nearby SF [25,26,37]. Furthermore, the occurrence of a homo-FRET process among the W67 residues present in the four KcsA channel subunits allowed for the calculation of intersubunit distances under different ionic conditions through the analysis of both steady-state and time-resolved fluorescence anisotropy [25]. These observations prompted us to prepare Y55W mutants of the NaK and NaK2K channels. As W67 in KcsA, the Y55 residue in the NaK and NaK2K channels is also located at the pore helix, and it was hoped that the study of the fluorescence properties of Y55W mutants could also provide useful information on cation-induced changes in the conformation of these channels. Such an expectation was further reinforced by a recent solid-state NMR paper on NaK-derived channels, reporting Y55 as a residue sensitive to the conformation of the SF [38].

Figure 2A,B show the normalized fluorescence emission spectra exhibited by the Y55W NaK and NaK2K mutant channels in the presence of high concentrations of either Na^+^ or K^+^. In all cases, the quantum yield of W55 was fairly high, near 0.3, similar to that shown by W67 in KcsA. Furthermore, similar to KcsA, the spectral shape and emission maxima of these channels were dependent on the ionic conditions used in the experiments. Thus, while the spectra in Na^+^ were very similar in both NaK and NaK2K channels, the spectrum of the NaK2K channel in K^+^ shows a large blue shift with respect to the NaK channel, which suggests that the specific interaction of K^+^ with the additional S1 and S2 sites introduced in the SF of the NaK2K channel causes a significant conformational change at the nearby pore helix, where the W55 reporter group is located, which cannot be resembled by the presence of Na^+^. Interestingly, the K^+^-induced blue shift in the NaK2K channel was larger by ~1 nm than that seen in KcsA [39], suggesting a more drastic change in the environment of the tryptophan reporter group in the NaK2K channel.

The steady-state characterization of the W55Y mutants of the NaK-derived channels also included the measurements of the average anisotropy in the presence of K^+^ or Na^+^. The samples were excited at 300 nm, as the fundamental anisotropy of tryptophan reaches its highest value (*r*(0)~0.3) at this wavelength [40], allowing for maximizing the dynamic range of the measurements [25]. As in the emission spectra from above, the steady-state anisotropy values revealed a dependence on the ionic conditions (Figure 2C,D). Moreover, a larger decrease in anisotropy is observed in the NaK2K channel in the presence of K^+^, compared to that in the NaK channel under similar conditions. It should be noted that in no case do the anisotropy values get near that of the fundamental anisotropy expected for a tryptophan residue (0.3), suggesting that, as in the case of W67 in KcsA, the W55 residues in these channels are involved in a homo-FRET process leading to a larger decrease in anisotropy than expected. Moreover, the fluorescence properties of W55 in the NaK and NaK2K channels yielded a Förster radius (*R_0_*) of ~12–14 Å (Table 1, see Section 4 for details), similar to that found in W67 KcsA (12 Å; [25]). Thus, according to FRET theory, a change in FRET efficiency would be expected when the ratio between the inter-fluorophore distances and the Förster radius (*R*/*R**_0_*) falls between 0.8 and 1.7 (0.8 < *R/R_0_* < 1.7 [25], Appendix A), which in this case equals 10 to 22 Å. This way, since direct inspection of the intersubunit (lateral) distances between W55 residues in the available high-resolution crystal structures of NaK and NaK2K channels yield values around 17–19 Å (Table 1), it should be concluded that changes in homo-FRET among the W55 residues should be suitable to detect conformational changes in the pore-helices and the nearby SF. To test this hypothesis, time-resolved fluorescence and anisotropy measurements were carried out. In the first place, the average fluorescence lifetimes obtained from the fluorescence intensity decay in each sample were calculated as in [25] and are given in Appendix A. No significant differences were found between the two mutant channels, either in Na^+^ or K^+^. Lifetime values for these W55 channels varied between 4.5 and 5.1 ns [25]. In the second place, Figure 3B,C show representative time-resolved fluorescence anisotropy decays of the two Y55W mutant channels in both Na^+^ and K^+^ buffers. The anisotropy decays in the presence of K^+^ are always faster than that in Na^+^ for the two channels, which is a clear indication that a homo-FRET process favored in the K^+^-bound form is taking place among the W55 residues. This is particularly noticeable in the case of the NaK2K channel, in which the anisotropy decay in K^+^ is much faster than that in Na^+^. This is indicative of a large K^+^-induced conformational rearrangement at the SF of the NaK2K channel upon the interaction of K^+^ with the four available K^+^ binding sites in the latter channel, which efficiently brings the four subunits closer.

As in W67 KcsA, the four-fold axis of symmetry of the NaK-derived channels and the square geometry delimited by the W55 residues (Figure 3A) allowed for the calculation of the homo-FRET rate constants by fitting Equation (3) to the time-resolved anisotropy decay data. These decays are described by the concurrence of two depolarization processes: (i) the energy transfer between the four subunits of the proteins, and (ii) the overall rotational tumbling of the protein:detergent complex. In this work, the rotational correlation times of these complexes were independently determined by labeling the C15 residue in the full-length NaK channel with N-1-pyrene maleimide, yielding a value of 40 ± 3 ns (mean ± S.D., *n* = 4) (see Section 4 and Appendix A), which was fixed through the fitting procedure of the time-resolved anisotropy decays. The homo-FRET rate constants depend on the sixth power of the distance separating the W55 fluorophores involved in the FRET process (Equation (4)), and therefore, they serve to determine intersubunit distances at the level of the W55 residues within the channel complexes. Table 1 summarizes such intersubunit distances in the Y55W NaK and NaK2K channels and compares them to those previously determined in the W67 KcsA mutant channel [25]. As expected from the identical number of four K^+^ binding sites at their SFs, the NaK2K and KcsA Trp-Trp intersubunit lateral distances in the presence of 100 mM K^+^ were very similar, near 15 Å, while in the NaK channel, with only two K^+^ binding sites at its SF, the Trp residues were ~18 Å apart. As to the Na^+^-bound forms, intersubunit distances were estimated as ~25 Å in both NaK and NaK2K channel proteins, in contrast to the ~18 Å determined previously in KcsA. It should be noticed, however, that the 25 Å value given above underestimates the intersubunit distances in the presence of Na^+^ since such distances are much larger than the Förster radius. The comparison of the intersubunit distances from above has two clear implications. First, it evidences a vast conformational change that both NaK-derived proteins experienced when K^+^ was replaced by Na^+^ within the pore, especially in the case of the NaK2K channel. Second, even though NaK2K and KcsA share the same signature sequence, the much shorter intersubunit distances observed in the Na^+^-bound form of KcsA compared to NaK2K reveal a much tighter SF conformation in the KcsA channel. This could be related to the ability of KcsA to adopt a collapsed, non-conductive SF conformation in the presence of Na^+^ [7,41,42], which has not been detected in the NaK and NaK2K proteins.

### 2.3. Cation Binding to the Y55W Mutants of NaK and NaK2K Channels

As in previous studies on the K^+^-selective KcsA channel, several intrinsic fluorescence-based assays have been used here to study Na^+^ and K^+^ binding to the NaK-derived channels. Figure 4 illustrates the first of such assays, in which the cation-dependent changes in the thermal denaturation of the tetrameric channel proteins (tetramer dissociation into subunits and partial unfolding) are monitored [43]. The data clearly show that K^+^, but not Na^+^, increases the thermal stability of the NaK-derived channel proteins in a cation concentration-dependent manner. Such a stabilizing effect of K^+^ is more markedly seen in the four K^+^ binding sites containing NaK2K channels than in the two K^+^ binding sites containing NaK channels. A simple two-state unfolding equilibrium was fitted to thermal denaturation curves, such as those shown in Figure 4, to obtain the midpoint temperature (*t*_m_) for thermal denaturation at the different cation concentrations (Figure 5) [44]. Again, it is clearly observed that increasing the K^+^ concentration results in a large thermal stabilization of the two channel proteins, while increasing the Na^+^ concentration does not have any effect. For this reason, we continued analyzing only K^+^ titration data to estimate the apparent dissociation constants of cation (K^+^) binding to these channels. The increase in the *t*_m_ observed at increasing concentrations of the ligand (K^+^) is directly related to the degree of K^+^ occupancy of the selectivity filters [45,46,47] and can be used to estimate the dissociation constant of the channel−K^+^ complexes (inset to Figure 5A). It should be noticed that this model accounts for the existence of a single set of ligand binding sites on the protein, and therefore, the observed goodness in the fit throughout the whole K^+^ concentration range should be interpreted as that; indeed, there is only a single set of K^+^ binding sites in either the NaK or the NaK2K channels. The estimated dissociation constants for such single binding events are in the submillimolar range for both channels (Table 2). These results came somewhat as a surprise because the SF of the NaK2K channel, in particular, has an identical sequence to that of KcsA and, consequently, a similar behavior between KcsA and the NaK2K channel was expected. However, this is not the case since KcsA was previously shown to exhibit two K^+^ binding events in the micromolar and millimolar range, respectively [23,45,48].

To reassure the K^+^ binding features of the NaK-derived channels reported above from thermal denaturation assays, additional K^+^ binding assays based on monitoring different experimental observables were carried out. Figure 6 illustrates the changes observed in the fluorescence emission spectra, steady-state anisotropy, and W55-W55 intersubunit lateral distances (calculated from the time-resolved anisotropy decays, see some examples in Appendix A) in the presence of increasing amounts of K^+^. Both NaK and NaK2K channels presented a similar trend during the titrations: the spectral center of mass shifted to lower wavelength values (blue-shift), the <*r*> decreased (due to an increase in the homo-FRET efficiency), and the intersubunit distances at the pore helices level became shorter at increasing K^+^ concentrations. These results suggest that when K^+^ interacts with its binding sites at the SF, the newly established ion-channel interactions stabilize the structure of the protein by bridging the four subunits closer together and burying the Trp residues in a more hydrophobic environment. A brief comment should be made in the case of the NaK channel, in which, after the main stabilization of the fluorescence emission spectra characterized by a blue-shift of ~2 nm, a small red-shift was detected at K^+^ concentrations above 10 mM (Figure 6A). Since this shift is not accompanied by any modification in the intersubunit distances, it should be related to a local modification of the environment around the W55 residues, probably a change in hydration around the fluorescent reporter. In fact, molecular dynamic simulations found four water grottos connecting with the vestibule of the NaK channel, near the SF and pore helices, that are involved in regulating ion permeation [36].

The main differences between the non-selective NaK and the K^+^-selective NaK2K channels refer to an almost two-fold larger change in all the spectroscopic observables in the NaK2K channel, which seems correlative to the double number of K^+^ binding sites available at its SF. Nonetheless, regardless of the observable parameter selected, all binding plots in Figure 6 present a sigmoidal-like behavior and are described by a single K^+^ binding event that is usually saturated at >1–2 mM of this cation, in agreement with the thermal denaturation studies. The apparent dissociation constants derived from these data are also given in Table 2. Again, in contrast to the KcsA channel, where the binding of the permeant cation is described by two consecutive binding events [23,45,48], the NaK and NaK2K channels seemingly transit just from a putative empty SF to a conductive conformation as the K^+^ concentration increases (the latter form is thoroughly described in the X-ray crystallographic structures). Interestingly, both in NaK and in NaK2K, in which the open conformations are stabilized by the Δ19 deletion, the affinities to bind K^+^ are almost two orders of magnitude higher than that detected in the equivalent open-inactivated conformation of KcsA [48].

As in the thermal denaturation experiments from above, the Na^+^ titration of the NaK-derived channels does not result in any changes in the fluorescence emission spectra, steady-state anisotropy, or time-resolved anisotropy decays (Appendix A), preventing describing Na^+^ binding in these channels. Therefore, it seems that the putative Na^+^-protein interactions involved in the Na^+^ permeation through these channels do not translate into an effective modification of the homotetrameric structure of the NaK and NaK2K proteins at least under the experimental conditions used in this study.

## 3. Discussion

In this work, we study the conformational dynamics at the pore region of single-Trp mutants of the non-selective NaK channel and its partly K^+^-selective derivative NaK2K. The Y55W mutation introduced in the pore helix of the NaK-derived channels does not substantially change their functional properties compared to their non-mutated counterparts. Thus, the W55 NaK channel continues to be fully non-selective for either Na^+^ or K^+^, while the W55 NaK2K mutant is partly selective for K^+^ and shows a larger conductance for this latter cation. Moreover, the two mutant channels behave as permanently open channels, i.e., with a very high P_o_. In addition to the patch-clamp measurements in excised membrane patches, the functional behaviors of these channels were also observed by the ACMA (9-amino-6-chloro-2-methoxyacridine)-fluorescence assay in liposomes, which essentially confirms that NaK2K allows a much higher flux of K^+^ than NaK. Therefore, the observed functional responses indicate that the Y55W mutants used in this work may provide an excellent opportunity to use W55 residues as reporters to monitor the conformational events taking place at the channel pore and their dependence on the type and concentration of cations in the media and on the number of ion binding sites at the SF.

As previously seen in the single-Trp W67 KcsA mutant channel, the analogous W55 residue behaved as an excellent reporter of its local environment and of the connection between the ion occupancy of the selectivity filter and the conformation of the pore helices. Particularly, the optimal photophysical properties (high quantum yield, blue-shifted fluorescence emission spectra, average lifetimes ~5 ns, and a Föster’s radius in the range of the inter-fluorophores distances) were derived in an efficient homo-FRET process that allowed for the calculation of the intersubunit distances in different ionic environments. The analytical framework used in the present study to calculate the energy migration rate constants (*k*_1_) implied the same assumptions previously taken in the W67 KcsA channel [25]: (i) pure homo-FRET, fully reversible since all four fluorophores are spectroscopically identical; (ii) isotropic FRET, mainly due to mix polarization of the two low-lying singlet excited states, ^1^L_a_ and ^1^L_b_, with nearly perpendicular transition moments [40,49] and the fast (vibrational) local internal reorientations of Trp residues; (iii) unpolarized emission of indirectly excited tryptophan residues; (iv) isotropic rotational diffusion of the protein-DDM complex, dominated by a long rotational correlation time of ~40 ns that was fixed throughout the fitting procedure.

The first conclusion from our studies is that NaK and NaK2K presented similar structural characteristics in the presence of Na^+^ but very different in the K^+^-bound complexes, i.e., they behaved with different conformational plasticity. Thus, while the presence of Na^+^ does not have much of an effect on the SF of either channel (no changes in the *t*_m_ or *R* parameters even at concentrations as high as 1 M Na^+^), the presence of K^+^ causes important dose-response changes, more noticeably on the four K^+^ binding sites containing NaK2K. In this case, the established K^+^-protein interactions bring the four channel subunits closer together to delineate a much narrower pore (“snug-fit” effect of K^+^), burying the Trp residues in a more hydrophobic environment and resulting in a larger increase in the stability of proteins against thermal denaturation. The adoption of different conformations by these channels depending on the ionic conditions was previously suggested from NMR [17,18,19] and cryo-electron microscopy studies [20], but not from certain X-ray crystallographic studies [14,15,16]. As suggested by others [19], such an apparent discrepancy could be related, in crystallography, to a single structural state favored within the crystal, whereas in solution, the data reflect a population-weighted average of all states present. In any case, our data clearly illustrates that the structures adopted in the presence of K^+^ or Na^+^ are clearly different from each other. Particularly, the W–W intersubunit distances change from more than 25 Å in the presence of Na^+^ in both channels to approximately 18 Å and 15 Å, respectively, for the NaK and NaK2K channels in the presence of K^+^. In other words, the extremely flexible SF of these channels adapts to either large or small ionic species so that the SF diameter becomes highly dependent on the ions contained within it. High pore flexibility is believed to be the basis for the lack of selectivity in poorly selective ion channels [17,18,50,51,52] and may as well be an important factor in defining selectivity in these NaK-derived cshannels.

The lack of significant structural effects of the addition of Na^+^ on the conformation of these channels may also be an important additional factor in defining channel selectivity, particularly in the NaK2K mutant. In the truly K^+^-selective channel KcsA, the somewhat similar K^+^-induced changes observed in the SF conformation are accompanied by a Na^+^-induced blockade, i.e., selectivity is achieved not only by allowing K^+^ passage through an SF with an appropriate conformation to accommodate dehydrated K^+^, but also by preventing Na^+^ from passing through the pore. This lack of Na^+^ currents (only achieved at non-physiological extreme positive voltages [53]) seems to be associated with the formation of a “collapsed” conformation of the SF in the presence of this cation, where there is a constriction at the S2 site caused by the reorientation of the G77 residue [6,7]. In contrast, in the NaK-derived channels, there are no signs of SF collapse caused by Na^+^, especially surprising in the case of the NaK2K mutant, which harbors the same signature sequence as KcsA. Indeed, we have not been able to detect any interaction between Na^+^ and these channels, in which the SF remains in a looser conformation linked to a W55–W55 intersubunit distance of at least 25 Å, likely allowing hydrated Na^+^ to diffuse freely through the channel pore. Such a passive role of the SF in Na^+^ passage seems reminiscent of that reported for Na^+^ channels, in which the large, fully hydrated or hemi-hydrated Na^+^ passes through the channel SF in an unrestricted manner, without causing anything comparable to the “K^+^-induced fit” phenomenon seen in K^+^ channels [54,55]. In the case of the non-selective NaK channel, previous NMR and MD studies performed with lipid membrane-reconstituted proteins found higher structural plasticity in the SF when there is Na^+^ in the media, finding a dose-dependent response and detecting undefined conformations and even asymmetric deformation of the SF. This high conformational plasticity of the SF and surrounding areas seem to be essential for efficient Na^+^ conduction in non-selective channels [19] and could, in turn, be associated with the inability of our system to effectively detect significant Na^+^ binding when working in DDM micelles.

In addition to the lack of ability of the SF of the NaK derivatives to adopt a collapsed conformation in the presence of Na^+^, as commented above, the quantitative analysis of K^+^ binding provides additional clues to partly explain the functional properties of these channels. In KcsA, two consecutive binding events with high and low affinity to bind K^+^ by the SF, respectively, were detected [45]:$$KcsA (vacant SF)⇄\mathrm{KcsA}\cdot \;K^{+}(collapsed SF)⇄\mathrm{KcsA}\cdot \;2K^{+}\left(conductive SF\right)$$

The first binding event presented a high affinity (micromolar K_D_ values), assigned to the crystallographic S1 and S4 sites (collapsed SF) and secure displacement of the competing non-permeant cations, while the second set of sites with low affinity (millimolar K_D_ values), contributed to by all S1 to S4 sites, favors cation dissociation and permeation. In contrast, in the NaK–derived channels, we have not been able to detect the high affinity set for K^+^ binding, and therefore, this could be an additional reason to explain the partial or no selectivity exhibited by these channels, as they would not be able to efficiently select K^+^ from other cations. A similar effect was observed in the non-inactivating E71A KcsA mutant channel, where the first K^+^ binding event is also absent, and a poor affinity for Na^+^ was detected compared to the WT protein [23,56,57]. Thus, similar to the E71A KcsA mutant, K^+^ binding to the NaK and NaK2K channels could be described by a single equilibrium:$$NaK or NaK2K(vacant SF)⇄\mathrm{NaK}\cdot {\;K}^{+}or NaK2K\cdot {\;K}^{+}\left(conductive SF\right)$$

In KcsA, as in many eukaryotic K^+^ channels, a C-type inactivation process takes place in the ms time range following channel opening. Initially, the structure of the inactivated state was associated with the collapsed conformation of the SF described by X-ray crystallography at low mM concentrations of K^+^ [58]. More recently, however, the inactivated state has been described by more subtle changes in the conformation of the SF and by a decrease in the affinity for K^+^ [48,59]. In principle, the high-open probability detected in excised patches of the NaK-derived channels seems indicative of a lack of an inactivation process. Moreover, neither a “collapsed” structure of the SF, nor a decrease in the affinity for K^+^, have been detected in these channels by any biophysical approximation (including those used in this work). Therefore, this is to conclude that the NaK-derived channels, in spite of having their intracellular gate stabilized in an open conformation, do not inactivate. In this respect, the moderately high affinity reported here (K_D_s in the submillimolar range) and also in a previous ITC (Isothermal Titration Calorimetry) study [60] seems more related to a conductive conformation of the pore rather than to a low-affinity inactivated state, provide a plausible explanation to such lack of inactivation, based on preventing K^+^ loss from the SF. In fact, the slow-type inactivation has long been associated with a loss of K^+^ from the selectivity filter [4,5], which explains why C-type inactivation is favored at low K^+^ concentrations in potassium channels [6,7,8,9], while a higher concentration of the ion prevents it [9,24,61,62,63]. These observations led to the “foot in the door” or the “ion depletion of the pore” hypothesis, which proposed that the presence of ions inside the SF is fundamental to stabilizing it in the conductive conformation [10,11,12] and thus, prevent inactivation.

Regarding the permeation process, our results show that while the non-selective NaK channel conducts Na^+^ or K^+^ with similar conductance, the addition of two extra binding sites in the NaK2K variant causes a partial selectivity for K^+^ and a much higher flux of K^+^ than that of Na^+^. Thus, the presence of four consecutive K^+^ binding sites influences not only the selective properties of these channels, but also the rate at which each cation is conducted. In fact, the stack of four binding sites in the NaK2K channel, as in KcsA, causes a concentration-dependent “snug-fit” conformational change in the SF in which K^+^ sites must be occupied successively as the ions traverse the SF pushed by the “knock-on” forces driving the ion flow [51]. Interestingly, the comparison between the functional properties of NaK2K and KcsA seems indicative that a proper signature sequence is not the only structural element required to define the permeation properties of these channels. In fact, the improved selectivity for K^+^ in the NaK2K channel is still far from that exhibited by the truly selective KcsA, in spite of having an identical SF sequence. The conclusion is, therefore, that in order to achieve K^+^ selectivity, having the correct sequence of amino acid residues at the SF is a necessary but not sufficient condition. In this respect, several authors have entertained the idea that the protein scaffold behind the SF, particularly the nearby pore helices, may also play an important role in defining ion channel selectivity [33,38,64]. Here, we do not provide any evidence for the involvement of any particular residue or domain in the protein scaffold that might be involved in such a process. However, we found particularly attractive a proposal based on the involvement of the so-called inactivation triad domain in KcsA, formed by the E71-D80-W67 residues that are connected to the SF by an H-bond interaction network that also includes water molecules [65]. As shown in Figure 1, the equivalent to the E71 residue is not present in the NaK-derived channels, and the absence of the resulting interaction network could perhaps cause increased conformational flexibility in the SF by which the NaK-derived channels become able to accommodate species as different in size and properties as the hydrated Na^+^ or the dehydrated K^+^. Nonetheless, as in NaK channels, many other eukaryotic voltage-gated K+ channels lack a glutamic acid residue at the position equivalent to E71 in KcsA, but still inactivate after the gating process. This fact highlights the complexity of the process and the importance of other scaffold residues around the SF as possible modulators of its conformation and of the functional properties of these membrane proteins.

In summary, Y55W mutants of the non-selective NaK and partly K^+^-selective NaK2K channels used in this work have proven to be a useful tool for exploring the conformational dynamics at the pore region of these channels in their interaction with either Na^+^ or K^+^. The first major conclusion is that these channels exhibit a remarkable pore conformational flexibility. Homo-FRET measurements in the presence of Na^+^ or K^+^ reveal a large change in W55-W55 intersubunit distances, enabling the SF to accommodate different species, thus, favoring poor or no selectivity. Depending on the cation, these channels exhibit wide-open conformations of the SF in Na^+^, or tight induced-fit conformations in K^+^, the latter of which is most favored in the four binding site-containing NaK2K channel. This conformational flexibility seems to arise from the lack of restricting interactions between the SF and the protein scaffold behind it. Such interactions, which vary among the K^+^ channels superfamily, appear as necessary elements to complement the proper signature sequence at the SF in fully defining the selectivity and permeation properties. Moreover, cation binding experiments provide additional clues to the poor or no selectivity of these channels. Indeed, compared to the K^+^ selective KcsA channel, the NaK-derived channels lack a high affinity K^+^ binding component and do not collapse in Na^+^. Thus, these channels cannot properly select K^+^ over other cations or reject Na^+^ by collapsing, as K^+^ selective channels do. Finally, these channels do not show C-type inactivation, likely because their submillimolar K^+^ binding affinities prevent an efficient K^+^ loss from their SF, thus favoring permanently open channel states.

## 4. Materials and Methods

### 4.1. Molecular Biology, Protein Expression, and Purification

The NaK construct in the pQE-60 expression vector was provided by Dr. Youxing Jiang (University of Texas Southwestern Medical Center, Dallas, TX, USA), and oligonucleotide-based, site-directed mutagenesis was then used to introduce the point mutation Y55W (NTZYTech, Lisboa, Portugal). A codon-optimized version of the NaK2KW55 (NaK D66Y, N68D, and Y55W) was custom synthesized (GenScript Biotech, Rijswijk, Netherlands) to reduce its guanine-cytosine content and subcloned into pET-28a. Protein sequences were:
NaKMAKDKEFQVL^10^FVLTILTLIS^20^GTIFYSTVEG^30^LRPIDALWFS^40^VVTLTTVGDG^50^NaK2KMAKDKEFQVL^10^FVLTILTLIS^20^GTIFYSTVEG^30^LRPIDALWFS^40^VVTLTTVGYG^50^NaKNFSPQTDFGK^60^IFTILYIFIG^70^IGLVFGFIHK^80^LAVNVQLPSI^90^LSNLVPRGSR^100^NaK2KDFSPQTDFGK^60^IFTILYIFIG^70^IGLVFGFIHK^80^LAVNVQLPSI^90^LSNLVPRGSR^100^NaKSHHHHHH^107^NaK2KSHHHHHH^107^

The C-terminal hexahistidine tagged NaK and NaK2K channels were overexpressed in *E. coli* M15 (pRep 4) and BL21 (λDE3)-competent cells [66] with the following modification. *E. coli* competent cells were transformed with the above constructs following standard heat-shock procedures and plated overnight on LB agar. A single colony was picked up and grown overnight at 30 °C in 100 mL of LB medium; 50 mL of this culture was diluted into 1 L of 2×YT medium and grown at 37 °C to exponential phase (at an absorbance at 600 nm of ~0.8). Protein expression was induced by 0.5 mM isopropyl β-D-thiogalactopyranoside (IPTG) at 30 °C for 3 h. Cells were pelleted and suspended in 100 mL of buffer (20 mM HEPES, pH 7.5, 0.45 M sucrose) containing one EDTA-free protease inhibitor mixture tablet (Roche), 0.4 mg/mL lysozyme, and kept on ice for 1 h. The mixture was sonicated on an ice bath using a Branson probe-type apparatus and centrifuged for 45 min at 100,000× *g*. Membrane proteins in this crude membrane pellet were solubilized in 20 mL of 20 mM HEPES, pH 7.5, 100 mM KCl, 50 mM imidazole, and 10 mM n-dodecyl β-D-maltoside (DDM; Calbiochem, San Diego, CA, US) for 2 h at room temperature. After centrifugation of insoluble remains (45 min at 100,000× *g*), the supernatant was incubated with Ni^2+^-Sepharose 6 fast flow resin (GE Healthcare, Chicago, IL, USA) overnight at 4 °C, placed into a column, and washed with 20 mM HEPES, pH 7.5, 100 mM KCl, 10 mM imidazole, and 1 mM DDM, until the absorbance at 280 nm was less than 0.01. The gel-bound protein was eluted using 3 mL of the previous buffer containing 500 mM imidazole. Protein-containing fractions were mixed and then concentrated using 10 kDa cut-off centrifugal filters (Millipore, Burlington, MA, USA). The tetrameric fraction was subsequently purified by using a Superdex 200 10/300 GL column in 20 mM Hepes, pH 7.0, 100 mM KCl, 1 mM DDM buffer and kept at 4.0 °C until use. Protein concentration was routinely determined from the absorbance at 280 nm, using a molar extinction coefficient of 8250 M^−1^cm^−1^ for NaKW55 and 9530 M^−1^cm^−1^ for the NaK2KW55 [67].

### 4.2. Ion Channel Reconstitution into Liposomes

NaK and NaK2K channels were reconstituted in asolectin liposomes for the activity assays, as previously reported [68]. Briefly, the required amount of lipid was dissolved in chloroform:methanol (2:1, by volume), and the solvents removed using a rotary evaporator and vacuum. The dried lipid film was resuspended at 20 mg/mL in 10 mM HEPES pH 7.0, 100 mM KCl and stored in liquid nitrogen. Before use, defrosted lipid suspensions were diluted to 5 mg/mL, then vortexed and sonicated to clarity. DDM solubilized protein channels at approximately 1 mg/mL were added drop by drop to the lipid solution while being vortexed to give a lipid-to-protein ratio of 100:1 by weight. The detergent was removed using Bio-Beads SM-2 (Bio-Rad laboratories), and, after discarding them, the reconstituted liposome suspensions were collected by centrifugation and finally suspended in 10 mM HEPES pH 7.0, 100 mM KCl. Samples were stored at −80 °C.

For patch-clamp experiments, 4 µL of the above-reconstituted liposomes were placed on a clean glass slide and dried overnight in a desiccator chamber at 4 °C and then rehydrated with 20 μL of 10mM HEPES (pH 7.0), yielding multilamellar giant liposomes after a few hours of rehydration.

### 4.3. Ion Channel Functional Measurements

Patch-clamp recordings were conducted on excised patches from asolectin giant liposomes containing NaK and NaK2K, as reported previously [68]. Recordings were obtained using an EPC-10 (Heka Electronic, Lambrecht/Pfalzt, Germany) patch-clamp amplifier at a gain of 50 mV/pA. The holding potential was applied to the interior of the patch pipette, and the bath was maintained at virtual ground. An Ag-AgCl wire was used as the reference electrode. Data were digitized at a sampling rate of 10 kHz or 40 kHz (when recording continuous pulses), low-pass filtered to 2 or 8 kHz (Bessel filter, HEKA amplifier, HEKA Elektronik GmbH, Reutlingen, Germany), respectively, and analyzed with Clampfit 10.3 (Molecular Devices, Axon Instruments, San José, CA, USA). All measurements were taken at room temperature (24 °C).

Continuous recordings were performed at +150 and −150 mV under symmetrical conditions with HEPES 10 mM pH 7, and 200 mM of either K^+^ or Na^+^. Reversal potentials were determined under bi-ionic conditions in order to obtain the permeability ratio of Na^+^ relative to K^+^, P_Na_^+^/P_K_^+^, from the Goldman–Hodgkin–Katz equation [69]:(1)$$E_{rev}=\frac{RT}{F}ln\left(\frac{P_{K^{+}}{\left[K^{+}\right]}_{o}+P_{{\mathrm{Na}}^{+}}{\left[{\mathrm{Na}}^{+}\right]}_{o}}{P_{K^{+}}{\left[K^{+}\right]}_{i}+P_{{\mathrm{Na}}^{+}}{\left[{\mathrm{Na}}^{+}\right]}_{i}}\right)$$
where *E**_rev_* is the calculated reversal potential (zero-current potential), *R* the gas constant, *T* the temperature (297 K), *F* the Faraday constant, [K^+^]_o_ and [K^+^]_i_ are the extracellular and intracellular K^+^ concentrations, respectively, and [Na^+^]_o_ and [Na^+^]_i_ refer to extracellular and intracellular Na^+^ concentrations, respectively.

Liquid junction potentials between the pipette and bath solutions were calculated by using Clampex 10.3 (Molecular Devices, Axon Instruments, San José, CA, USA) and routinely corrected. To measure the reversal potential of the channel currents under bi-ionic conditions, the holding potential was stepped to different voltages (10 mV steps), and the null current potential for each seal was the x-axis intersection point of the plotted i/v values.

The addition of the classic potassium channel blocker TPeA (tetrapentyl ammonium) at 100 µM to the bath was used to ensure that recorded currents are from channels oriented with their external side facing the bath solution.

The fluorescence-based flux assay adapted from [31] was also used as an additional functional test in these samples. Briefly, reconstituted asolectin liposomes were thawed at 37 °C for 30 min, then sonicated in a bath for one minute to eliminate aggregates. Immediately before the flux assay, liposomes were diluted in HEPES 10 mM pH 7, NMDG 100 mM, to create a gradient of K^+^, since the internal buffer is HEPES 10 mM pH 7, KCl 100 mM. Then, the fluorescent 9-amino-6-chloro-2-methoxyacridine (ACMA) (Sigma-Aldrich, Madrid, Spain) probe was added to the liposomes to a final concentration of 1 µM. The fluorescence of this probe was continuously measured at an excitation and emission wavelength of 410 and 490 nm, respectively, in an SLM-8000C spectrofluorimeter (SLM, Urbana, IL, USA). Once the fluorescence signal is stabilized, the proton ionophore carbonyl cyanide m-chlorophenylhydrazone (CCCP) (Sigma-Aldrich, Madrid, Spain) was added at a final concentration of 1 µM to initiate the K^+^ efflux through the reconstituted channel, due to the counterbalanced influx of H^+^. This process is monitored through the ACMA protonation, which quenches its fluorescence and avoids its permeation through the membrane. After ion and proton flux reached an equilibrium, available liposome volume was measured by the addition of valinomycin, which provides a channel-independent path for K^+^ conduction. Data shown are normalized to the maximal and minimal fluorescence measured when the CCCP and valinomycin were added, respectively.

### 4.4. Steady-State Fluorescence Spectroscopy

Steady-state fluorescence measurements were performed on an SLM-8000C spectrofluorometer using 0.5 cm pathlength quartz cuvettes (Hellma, Müllheim, Germany) at room temperature. The fluorescence emission spectra were acquired at λ_exc_: 295 nm and corrected with an appropriate blank. The results are expressed in terms of the fluorescence spectral center of mass (intensity-weighted average emission wavelength, <λ>) as defined in [39]. The steady-state fluorescence anisotropy, <*r*>, was measured at 340 nm using an excitation wavelength of 300 nm to maximize the dynamic range of the time-resolved anisotropy measurements [25]. Ten measurements were conducted for each sample, and two independent samples were used to calculate average steady-state anisotropy values.

The Förster radius, *R*_0_, or critical distance at which the energy transfer efficiency is 50% for an isolated donor–acceptor pair, was calculated for the W55 residues in both channels using the following relationship [70]:(2)$$R_{0}\;\left(Å\right)=0.2108\;{\left[\kappa^{2\;}n^{-4}\;\Phi_{D}\;J\left(\lambda \right)\right]}^{\frac{1}{6}}$$
where the orientation factor, *κ*^2^, and the refractive index of the medium, *n*, were assumed to be 2/3 (i.e., the dynamical isotropic limit value) and 1.6, respectively. The donor quantum yield, *Φ_D_*, for the W55 residue was calculated using a reference solution of *N*-acetyl-*L*-tryptophanamide (NATA) in water (*Φ* = 0.14; [71]). These latter values were usually 0.27–0.3. The spectral overlap integral, *J*(λ), was calculated using the absorption and normalized fluorescence emission spectra of each sample.

### 4.5. Fluorescence Monitoring of Cation Binding

Thermal denaturation of DDM-solubilized proteins in the presence of increasing amounts of the different cations was performed in a Varian Cary Eclipse spectrofluorometer (Agilent) by monitoring the temperature dependence of the protein intrinsic fluorescence emission at 340 nm after excitation at 280 nm, as in [43]. Batteries of samples were prepared from the dilution of the protein stock to 2 µM final protein concentration in 20 mM Hepes, pH 7.0 buffer containing 1 mM DDM, and the corresponding cation being tested. The midpoint temperature of the thermally-induced protein denaturation process in the different samples (*t*_m_) at the different cation concentrations was calculated from the thermal denaturation curves by fitting a two-state unfolding model to the data, assuming a linear dependence of the pre- and post-transition baselines on temperature, according to [44].

The binding of K^+^ to the NaK and NaK2K single-Trp mutants was also monitored by the K^+^ concentration-dependent changes on the <λ>, <*r*>, or *R* fluorescence parameters. Such changes presented a sigmoidal behavior associated with a single equilibrium from an empty or vacant SF to a conductive form. The apparent K_D_ values were then calculated using a two-state model, assuming a cooperative process that is described by an empiric Hill function [26].

### 4.6. Time-Resolved Fluorescence Spectroscopy

Time-resolved fluorescence and anisotropy measurements with picosecond resolution were obtained using the single-photon timing (SPT) technique, as described elsewhere [72]. The fluorescence and anisotropy decays of the 6–8 µM DDM-solubilized NaK and NaK2K samples (λ_exc_ = 300 nm; λ_em_ = 345 nm), were acquired using a frequency-doubled Rhodamine 6G laser and analyzed as previously with the TRFA software (Scientific Software Technologies Center, Minsk, Belarus), which uses a non-linear least squares regression method based on the Levenberg–Marquardt algorithm. The usual statistical criteria, namely a reduced χ^2^ < 1.2 and a random distribution of weighted residuals and autocorrelation plots, were used to evaluate the goodness of fits. The amplitude-weighted average lifetime <τ>_1_ and the intensity-weighted average lifetime <τ>_2_ were calculated as previously described [27].

### 4.7. Determination of the Rotational Correlation Time of the Channel-DDM Complexes

To determine the overall rotational correlation time ($\phi_{g}$) of the detergent-solubilized NaK derivatives, the native C15 position of the full-length NaK channel was chemically modified with the long-lifetime fluorescent probe *N*-(1-pyrene) maleimide, as previously performed in the KcsA channel [25]. Both the fluorescence intensity and anisotropy decays of the pyrene-labeled mutant solubilized in DDM were measured in the presence of either 100 mM K^+^ or 100 mM Na^+^ (Appendix A) (λ_exc_ = 335 nm, λ_em_ = 400 nm), and analyzed using the TRFA software. As expected, the intensity-weighted mean fluorescence lifetime of the detergent-solubilized pyrene-labeled protein was very long (<τ>_2_~50 ns). The anisotropy decays of this pyrene-conjugated mutant were found to be essentially independent of the experimental conditions used (Appendix A). The dominant long rotational correlation time, $\phi_{g}$~ 40 +/− 3 ns was assigned to the overall tumbling of the tetrameric NaK-DDM complex in buffer solution.

### 4.8. Analysis of the Time-Resolved Anisotropy Decays to Determine the W55-W55 Intersubunit Distances

The square arrangement of the W55 residues within the homotetrameric structure of the single-Trp NaK derivatives allowed for the calculation of the intersubunit (lateral) distances, *R*, using the same analytical framework described for the W67 KcsA mutant channel [25]. Since all four W55 residues are located in identical environments, they were considered spectroscopically equivalent. The homo-FRET lateral rate constants, *k*_1_, were calculated from the time-resolved anisotropy decays as:(3)$$\left(t\right)=\frac{r\left(0\right)}{4}\left[1+\exp\left(-4k_{1}t\right)+2\exp\left(-\frac{9}{4}k_{1}t\right)\right]\cdot \exp\left(-t{/\phi }_{g}\right)$$
where *r*(0) is the initial anisotropy (time = 0), and $\phi_{g}$ is the rotational correlation time of the NaK-DDM complex in solution. The latter parameter was independently determined (see above-section) and fixed in all the fitting procedures. Since this model assumes that FRET is isotropic, then lateral distance R can be calculated by:(4)$$k_{1}=\frac{1}{\tau }{\left(\frac{R_{0}}{R}\right)}^{6}$$
where *τ* is the intensity-weighted mean fluorescence lifetime, and *R*_0_ is the Förster radius.

Unless otherwise indicated, values given in the text are the mean ± the standard deviation (S.D.) of three independent experiments.

## Data Availability

Data is contained in the article.

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
