# Peer review of "Molecular Events behind the Selectivity and Inactivation Properties of Model NaK-Derived Ion Channels"

_ijms, 2022, doi:10.3390/ijms23169246_

Round 1

Reviewer 1 Report

I would like to congratulate the authors on a thorough and well designed study. The techniques utilized are challenging but well executed. The data are well analyzed and presented.  The study is very well described in the text and the conclusions drawn are appropriate. It is rare to review work and not have substantive changes to require before recommending publication.  I have neither substantive nor minor changes to suggest in this case.  Excellent work!

Reviewer 2 Report

Giudici, Renart and co-workers examine the conformational dynamics of NaK and NaK2K channels at the pore region by using Y55W mutants of their channels that are reconstituted in asolectin liposomes.  The experiments were performed by using the patch-clamp technique (applied to patches excised from channels-containing liposomes), fluorescence-based flux assay and so on.  Based on analyses of several data obtained here with reference to published data of KcsA channel, the authors concluded that the NaK and NaK2K channels exhibit a remarkable pore conformational flexibility.  Of so many types of K+ channels that exit, the NaK and NaK2K channels are minor ones and the conformational dynamics of the channels examined in this study are detailed.  Therefore, this study may not be of much interest to the general reader who is interested in K+ channels.  There are many minor points that may be useful to amend this manuscript, as follows:

1.     Eqs. 1 and 2 given in Materials and Methods section are not mentioned in Results section.  Please amend this point.

2.     Line 41: not “have” but “has”?

3.     Line 62: it is unnecessary to define “SF” twice (see line 45).  Please amend this point.

4.     Line 120: “recon-stituted” should be “reconstituted”.

5.     Line 142: please give the number of seals examined but not “at lease three seals”.

6.     Line 175 “figure 1” should be “Figure 1”.

7.     Line 196: not “detected” but “detect”.

8.     The first paragraph on page 5: the I-V curve of Fig. 1 does not show inward rectification.  Is not it necessary to show the I-V curve exhibiting inward rectification?  How many patches were used in the experiment to investigate reversal potential?  “around +19 mV” and “about 0.4” (in line 190) should be shown as average and SD.

9.     Line 214: “that that” should be “that”.

10.  Line 234: it may be better for DDM to be defined in this line but not in lines 648 and 649.

11.  Table 1: how many experiments did the mean and SD shown in this table come from?  This should be written in the footnote of this table.

12.  Line 307: not “time” but “times”?  Please check English.

13.  Line 309: it should be written how many experiments “40 ± 3” was obtained from.

14.  Line 324: not “have” but “has”?  Please check English.

15.  Page 8: the authors should use either Equation or Eq. (page 20).

16.  The second paragraph on page 9: was Fig. 5 obtained by performing the experiment as shown in Fig. 4 three times (see line 377)?  Please make this point clear.

17.  Table 2: this table seems to show a range of the KD values obtained.  Did all of this data come from three experiments?  What is the SD value of the average value given here?  Please reply to these questions.

18.  Line 398: not “interact” but “interacts”?

19.  Line 451: it may be better for ACMA to be defined in this line but not line 713.

20.  Line 477: not “1M” but “1 M”.

21.  Line 490: is English in this line OK?  Please check the sentence in this line.

22.  Line 495: is “the ions within” OK?  Please check English.

23.  Line 505: not “[54]” but “[54])”?  Please check this matter.

24.  Line 559: please expand “ITC”.  Not “seem” but “seems”?

25.  Line 587: is “attractive a proposal” OK?  Please check English.

26.  Line 589: is “include” OK?  Please check English.

27.  Line 612: not “.. poor o no ..” but “.. poor or no ..”?  Please check English.

28.  Line 640: please use either “mL” or “ml” throughout the text.  oC” should be changed to “oC” throughout the text.

29.  Lines 688 and 697: was room temperature 24 oC?  Please reply to this question.

30.  Line 706: “bαth” should be “bath”.

31.  Line 724: not “showed” but “shown”?

32.  Line 733: not “were” but “was”?

33.  Lines 746 and 747: is “was using” OK?  Not “.. [72].” but “..[ 72]).”?

34.  References: is “Science (80-.  )” in line 839 OK?  Is “IUCrJ” on line 910 OK?  “+” in “K+” should be superscript throughout references.  Moreover, please check all of the references whether they are cited correctly.

35.  There appear to be more mistakes than pointed out above.  Please check the manuscript very carefully.
